# Serology Assays Used in SARS-CoV-2 Seroprevalence Surveys Worldwide: A Systematic Review and Meta-Analysis of Assay Features, Testing Algorithms, and Performance

**DOI:** 10.3390/vaccines10122000

**Published:** 2022-11-24

**Authors:** Xiaomeng Ma, Zihan Li, Mairead G. Whelan, Dayoung Kim, Christian Cao, Mercedes Yanes-Lane, Tingting Yan, Thomas Jaenisch, May Chu, David A. Clifton, Lorenzo Subissi, Niklas Bobrovitz, Rahul K. Arora

**Affiliations:** 1Cumming School of Medicine, University of Calgary, Calgary, AB T2N 4N1, Canada; 2Institute of Health Policy Management and Evaluation, University of Toronto, Toronto, ON M5T 3M6, Canada; 3Wyss Institute for Biologically Inspired Engineering, University of California Berkeley, Berkeley, CA 02115, USA; 4Faculty of Science, University of Calgary, Calgary, AB T2N 1N4, Canada; 5Temerty Faculty of Medicine, University of Toronto, Toronto, ON M5S 1A8, Canada; 6COVID-19 Immunity Task Force, McGill University, Montreal, QC H3A 0G4, Canada; 7Department of Epidemiology & Center for Global Health, Colorado School of Public Health, Aurora, CO 80045, USA; 8Institute of Biomedical Engineering, University of Oxford, Oxford OX3 7DQ, UK; 9World Health Organization, 1211 Geneva, Switzerland; 10Department of Critical Care Medicine, University of Calgary, Calgary, AB T2N 4N1, Canada

**Keywords:** serological assay, seroprevalence, performance, sensitivity, specificity, evaluation, validation

## Abstract

Background: Many serological assays to detect SARS-CoV-2 antibodies were developed during the COVID-19 pandemic. Differences in the detection mechanism of SARS-CoV-2 serological assays limited the comparability of seroprevalence estimates for populations being tested. Methods: We conducted a systematic review and meta-analysis of serological assays used in SARS-CoV-2 population seroprevalence surveys, searching for published articles, preprints, institutional sources, and grey literature between 1 January 2020, and 19 November 2021. We described features of all identified assays and mapped performance metrics by the manufacturers, third-party head-to-head, and independent group evaluations. We compared the reported assay performance by evaluation source with a mixed-effect beta regression model. A simulation was run to quantify how biased assay performance affects population seroprevalence estimates with test adjustment. Results: Among 1807 included serosurveys, 192 distinctive commercial assays and 380 self-developed assays were identified. According to manufacturers, 28.6% of all commercial assays met WHO criteria for emergency use (sensitivity [Sn.] >= 90.0%, specificity [Sp.] >= 97.0%). However, manufacturers overstated the absolute values of Sn. of commercial assays by 1.0% [0.1, 1.4%] and 3.3% [2.7, 3.4%], and Sp. by 0.9% [0.9, 0.9%] and 0.2% [−0.1, 0.4%] compared to third-party and independent evaluations, respectively. Reported performance data was not sufficient to support a similar analysis for self-developed assays. Simulations indicate that inaccurate Sn. and Sp. can bias seroprevalence estimates adjusted for assay performance; the error level changes with the background seroprevalence. Conclusions: The Sn. and Sp. of the serological assay are not fixed properties, but varying features depending on the testing population. To achieve precise population estimates and to ensure the comparability of seroprevalence, serosurveys should select assays with high performance validated not only by their manufacturers and adjust seroprevalence estimates based on assured performance data. More investigation should be directed to consolidating the performance of self-developed assays.

## 1. Introduction

Serosurveys have been foundational to emergency pandemic surveillance and evidence-guided public health policy during the COVID-19 pandemic. These studies help map the true extent of SARS-CoV-2 infection, indicators of population humoral immunity, and other measures of disease risk [1]. Serological assays, the laboratory tools for detecting antibodies produced after SARS-CoV-2 infection or vaccination, are a critical methodological step in serosurvey design and result interpretation. In response to expanding demand for serosurveys, many SARS-CoV-2 serological assays were developed, mobilized, and adopted since the beginning of the pandemic.

The breadth of available serological assays since the beginning of the pandemic is large and diverse, with over hundreds of serological assays currently commercially available. Most serological assays target antibodies against the spike (S) and/or nucleocapsid (N) proteins [2] of the SARS-CoV-2 virus and detect a variety of antibody isotypes (IgG, IgM, IgA, or all—Total Ab). To date, several types of analyte binding methods and virological techniques have been applied to SARS-CoV-2 serology—the most common being neutralization assays, lateral flow immunoassays [LFIAs], immunofluorescence assays [IFAs], enzyme-linked immunosorbent assays [ELISAs], and chemiluminescence assays [CLIAs].

An important consideration during serosurvey study design is assay performance. Assay performance has direct consequences on the validity of a study, where the sensitivity (Sn.) and specificity (Sp.) reflect whether a given seroprevalence result is accurately reflective of the sample group’s true antibody positivity. Sn. and Sp. are not fixed properties of an assay—they are dependent on the panel of samples they were tested with. Manufacturers, third-party sources, and other independent groups conduct performance evaluations on the Sn. and Sp. of assays to ensure the reliability and comparability of seroprevalence results. These evaluations use panels with different compositions of samples, some of which are likely to produce high estimates. Thus, the evaluation performance of assays varies considerably. Recently, a review compared serological assay performance against RT-PCR results for 58 studies [3]. The authors found that among ELISAs, CLIAs, and LFIAs, the pooled assay Sn. and Sp. ranged from 75–91% (Sn.) and 92–100% (Sp.). This broadly varying assay performance raises the concern that SARS-CoV-2 seroprevalence estimates may be biased by imperfect or inconsistent assay performance, especially in cases where no statistical adjustments are made to account for test performance. 

Validation from different sources is often in disagreement and results in varied intra-assay performance data especially compared to manufacturer-certified evaluations, as supported by several head-to-head laboratory assay comparison studies [4,5,6,7,8]. Commercial assays constitute the vast majority of assays used in serosurveys, and manufacturers of these commercial assays self-certify their testing products with in-lab evaluations [9]. Such evaluations were usually done in the early pandemic using small true positive samples drawn from patients with confirmed symptomatic COVID-19 and no co-infection of other viruses [10]. The lack of endemic samples representing the demographics and endemic pathogens in a study area introduces spectrum bias [11]. There is also a lack of standardization between the methodology for manufacturer evaluations, and key factors such as the time of post-symptom onset that sampling was performed vary. 

There is uncertainty in the extent to which mis-specified assay performance will introduce bias to results in unadjusted and adjusted seroprevalence estimates. This issue is further exacerbated by the discordant validation data between sources and the unavailability of third-party evaluations for certain assays. For this reason, there is a need to synthesize assay performance data for use in both the design and interpretation of serosurveys. In particular, how these sources of validation data differ and what Sn. and Sp. of an assay are needed to minimize bias in seroprevalence estimates given the true background prevalence. These results have important implications for public health policy and resource mobilization through the interpretation of seroprevalence data: especially critical for the future course of the pandemic and advising serosurveillance for future infectious disease threats.

Our group maintains a living systematic review of SARS-CoV-2 seroprevalence [12]. We sought to (1) describe features and usage of serological assays, as well as the implementation of testing algorithms employing multiple tests in SARS-CoV-2 serosurveys during the COVID-19 pandemic; (2) comprehensively compare the performance of these assays across manufacturers, third-party reference labs, and independent investigator evaluations; and (3) quantitatively assess the influence of assay performance on seroprevalence estimates. To our knowledge, this is the first large-scale evaluation of discrepancies between validation sources and intra-assay performance for serological assay targeting SARS-CoV-2 antibodies.

## 2. Materials and Methods

This study is registered as a part of a living systematic review of global SARS-CoV-2 seroprevalence studies in PROSPERO (CRD42020183634 [12]), which is also accessible on the open-access web dashboard, SeroTracker [13]. Detailed methods and results from this review have previously been published [14,15]. 

### 2.1. Data Sources and Search Strategy

We developed a search strategy intending to be as thorough as possible in comprehending the immunoassays utilized in seroprevalence studies. The search strategy identified published literature, and preprints were created in collaboration with a health sciences librarian. We sought to reduce any potential publishing bias by adding a range of sources besides peer-reviewed publications, including institutional reports, media sources, and grey literature. Experts who collaborated with us and the users of the SeroTracker website recommended grey literature [14]. All identified articles were recorded in the SeroTracker database. This is a database that contains the most complete source of seroprevalence research ever made available. From 1 January 2020 to 19 November 2021, we searched for articles on Medline, EMBASE, and Web of Science preprints on Europe PMC. Our secondary search included Google News, articles submitted to SeroTracker.com, and studies submitted to us by expert recommendations. Two reviewers independently screened titles/abstracts and full texts. Data were extracted and critically appraised in duplicate [16]. 

### 2.2. Inclusion and Exclusion Criteria

We included all SARS-CoV-2 seroprevalence studies on humans which reported a sample size, sampling date and locale, and prevalence estimate. We excluded studies conducted only on people with SARS-CoV-2 infection or vaccination and online public dashboard estimates that were not associated with a defined serology study [15]. We adapted an automated appraisal tool based on the Joanna Briggs Institute critical checklist to evaluate the risk of bias in included seroprevalence studies [17]. Full details of the assessment process can be accessed from this preprinted work [16].

### 2.3. Serological Assay Data Extraction

We extracted all data for serological assays to develop an independent database linked to the master seroprevalence study database. It included assay-related information reported in individual seroprevalence studies. For each assay, we identified product name, manufacturer or developer, country, WHO geographical region of development, antibody isotypes detected (IgG, IgM, IgA, total Ab), test type (ELISA, LFIA, IFA, CLIA, neutralization assay, others; Appendix A), antibody target (Spike, Nucleocapsid, others), multiplex detection (detecting more than one antibody targets), time to result (Rapid Diagnostic Tests [RDT]/non-RDT), and test Sn. and Sp. as reported by manufacturers or developers. For commercial assays, we validated and complemented details on assays using reference links provided by authors. These links directed us to manufacturer’s websites or user’s guides which contained detailed information on the given assay. For self-developed assays, reference links pointed to the original research article with comprehensive development details.

Many studies cited serological assay validation results to corroborate the performance of the assay they selected. However, given that the testing environment, validation procedure, and reference panel varied across groups conducting validation, we categorized assay validation as either (1) third-party lab validation or (2) independent group field validation. We linked commercial assays with their performance in five large third-party lab performance evaluations and defined these as third-party lab validations. These five labs conducted large-scale head-to-head evaluations under controlled and reproducible conditions, including NRL (WHO sponsored [4]), the US FDA [5], Netherland CIDC [6], The Doherty Institute [7], and FIND Diagnostics [8] (Appendix A). Independent field validation results were defined as performance validation data extracted from individual seroprevalence studies. These studies reported pretest results with a smaller sample in addition to population prevalence. Where available, assays’ Sn. and Sp. for all isotypes and total antibodies were extracted from third-party lab evaluations and independent evaluations. The WHO has set performance criteria for the emergency use of Sn. >= 90.0% and Sp. >= 97.0% [18]. We applied these thresholds to categorize commercial assays based on performance in manufacturer, third-party, and independent evaluations.

Evaluation data was not very available for self-developed assays as for commercial assays, in the assay description of which concentrated on the steps of developing such an assay with performance matrices provided randomly. Therefore, corresponding performance analysis was not conducted for self-developed assays.

### 2.4. Analysis

Data extraction, cleaning, and management were performed in a collaborative data collection platform (Airtable.com, accessed on 20 November 2022). Data analysis was performed using R 4.0.2 [19]. We first summarized basic study characteristics, seroprevalence estimates, and serological assay features stratified by the WHO region at the study level.

At the assay level, we described the distribution of test usage, initial adoption, test type, region of development, test features, test evaluation states, and eligibility for emergency use by commercial and self-developed assays. We collected Sn./Sp. data to show the difference in reported performance for the top 50 assays and the top 20 assays by evaluation sources (manufacturers, third parties, and independent groups). The median Sn. and Sp. values for the top 50 assays were extracted from three evaluation sources and plotted on a panel against the WHO criteria. Bland–Altman plots were created to compare manufacturer-reported Sn./Sp. with a third party’s lab and independently evaluated Sn./Sp. in pairs.

For studies that used a testing algorithm involving multiple assays, we examined the combination of assays used (commercial/self-developed), how results from assays were combined (e.g., either test positive for a specimen to be positive vs. both tests positive), and whether the study reported a combined Sn. and Sp. for the testing algorithm. Many studies used multiple assay testing algorithms and also reported seroprevalence derived from using individual assays on the same set of samples. For these studies, we generated another set of Bland–Altman plots to show the discrepancy of estimates between testing algorithms. Seroprevalence estimates given by multiple assay algorithms and seroprevalence given by individual assays were compared in pairs.

### 2.5. Modeling Analysis

In examining whether assay performance differs by evaluation sources, we developed separate mixed-effect beta regression models for Sn. and Sp. with random effects specified for individual serological assays. Given that data with high heterogeneity, a diagonal heterogeneous variance–covariance structure was finally selected when estimating the assay performance by evaluation source. Assay features of isotype, test type, antibody targets, multiplex detection, and time to result were fitted as covariates to adjust outputs. Raw log odds obtained from models were converted to percentage for ease of interpretation. Difference in performance matrix against manufacturer values with 95% Confidence Interval (95% CI) by evaluation sources was derived using bootstrapping with 10,000 iterations. This modeling analysis enables us to determine discordance between evaluation sources and how inherent assay features may affect performance metrics.

We then performed a simulation. We simulated 1000 scenarios in which observed seroprevalence ranged from 0.0–99.9%. We adjusted assay performance [20] on observed prevalence to answer the third question we asked—to what extent a misreported assay performance value will bias the adjusted estimates from the ‘true’ prevalence estimate. The precise prevalence estimate intervals were defined by specifying error levels at ±5% [21] to the true prevalence. We simulated adjusted seroprevalence for assays at three accuracy levels—(1) high: Sn. = 95.0%, Sp. = 99.0%; (2) good: Sn. = 90.0%, Sp. = 97.0%; (3) moderate: Sn. = 87.0%, Sp. = 90.0%, with different levels of error of performance misspecification.

## 3. Results

### 3.1. Included Studies

We screened 72,799 titles and abstracts and 4876 full texts published between 1 January 2020, and 19 November 2021. This represents the pre-booster vaccine time window before Omicron where most qualitative tests were introduced. We extracted data from 2069 articles—262 of these were identified as preprints, overlapped by subsequent full articles. A total of 1807 serosurveys were included for final analysis (Appendix A). 

### 3.2. Assay Use in Seroprevalence Studies

Among these 1807 serosurveys, 80.7% of studies used a single serological assay (73.1% commercial assays, 18.2% self-developed assays, 8.7% unable to specify), while 19.3% used a testing algorithm involving multiple assays (Appendix A); 248 adjusted seroprevalence estimates for assay performance. Overall, global usage of commercial serology assays follows a power-law distribution, with the top 25 assays accounting for 67.0% of total commercial assay use in seroprevalence studies (Appendix A) and the top 50 assays accounting for 91.4% of use.

### 3.3. Characteristics of Identified Assays

Among 1807 serosurveys, we identified 192 commercial serology assays and 380 self-developed serology assays (Table 1). A full list of identified commercial serology assays can be found in Appendix A. Of the 192 identified commercial assays, 31.3% were ELISAs, 39.1% were LFIAs, 15.6% were CLIAs, 2.6% were IFA assays, and 15.6% were other types or not able to specify (Table 1). Of the 380 studies using self-developed assays, most used ELISAs (68.7%, Table 1). Product information was limited for many assays, most notably LFIAs: up to 32.6% and 42.6% of studies did not mention details about targeted antigen(s) and antibody isotypes, respectively. A total of 45.0% of studies using self-developed assays used multiplex detection to recognize multiple antibody targets. RDTs (types including LFIA, and IFA) accounted for 53.6% (103/192) of all commercial assays, while only 4.5% (17/192) of self-developed assays were developed as RDTs. 

### 3.4. Reporting of Assay Performance

Manufacturer data could be searched from publicly available online sources or manufacturer-led research papers for 91/192 (47.4%) commercial assays; 61.5% of these were subsequently either assessed in the five third-party evaluations or independent group evaluations (Table 2). Based on manufacturer data, the mean Sn. was 97.8 (95% CI: 93.9–100)% and the mean Sp. was 99.7 (95% CI: 97.8–100)%; 55/192 (28.6%) met the 90.0% Sn. and 97.0% Sp. WHO criteria for emergency use (Figure 1 and Figure 2); of the 50 most frequently used assays, 76.9% met the WHO criteria. In contrast, only 46.1% and 53.7% met these criteria based on third-party and independent evaluations, respectively (Figure 2).

CLIAs demonstrated higher and more reliable performance across all three evaluation sources than ELISAs, LFIAs, and IFAs among the top 50 assays (Figure 1 and Appendix A). The pairing comparison of manufacturer-reported figures of merit against five third-party lab and independent group evaluations indicated manufacturers systematically overstated the Sn. and Sp. of the assays they developed (Appendix A). After adjusting for assay features, Sn. and Sp. were considerably lower by 1.0% (95% CI: 0.1–1.4)% (*p* = 0.289) and 0.9% (95% CI: 0.9–0.9)% (*p* < 0.001) according to third parties and by 3.3 (95% CI: 2.7–3.4)% (*p* = 0.001) and 0.2 (95% CI: -0.1, 0.4)% (*p* = 0.247, Table 2) according to independent evaluations.

We conducted a simulation to examine the impact of incorrect Sn. and Sp. estimates on estimated seroprevalence, using a threshold of ±5% between true and adjusted prevalence to define substantial effects. Falsely specifying Sn. 5% higher than its true value will not affect population prevalence estimates for any assay with a higher than moderate performance (Sn. >= 80%, Sp. >= 87%). However, if Sn. is falsely specified by 10% higher and Sp. by 3% higher, population prevalence estimates are inaccurate for true prevalence below 18.3% or above 38.7% (assays with moderate performance), or inaccurate for true prevalence below 17.5% or above 41.5% (assays with good performance, i.e., Sn = 90%, Sp = 97%). Falsely specifying assay Sn. 10% lower and Sp. 5% lower than their true values lead to substantial deviations between estimated and true population seroprevalence at all seroprevalence values (Figure 3a–c, Appendix A).

### 3.5. Multiple Test Combinations

A total of 349/1807 (19.3%) studies employed a testing algorithm that used more than one serological assay (Appendix A). Most studies (254/349 (72.8%)) used a combination of the self-developed test(s) with self-developed test(s) and employed multiple laboratory-based (i.e., non-RDT) assays (267/349 (76.5%)). Concerning antibody targets, 152 (43.5%) studies combined spike and nucleocapsid-targeted assays, while spike–spike assay combinations were observed in 121/349 (34.7%) studies.

Of the 349 multiple-testing studies, 42.4% of these tested the same sample on multiple assays concurrently (“parallel testing”); among these, 68.2% defined seropositivity as a positive result on at least one assay, and 31.8% defined this as a positive result on all assays. A total of 31.8% used one assay first for screening, followed by another for confirmation (“sequential testing”). While having the combined Sn. and Sp. for a testing algorithm is important to interpret seroprevalence estimates, this was only reported in 9.5% of seroprevalence studies using multiple testing algorithms (Appendix A). A subset of samples from 167 studies tested on parallel or sequential multiple algorithms were identified to interpret seroprevalence estimates derived from these algorithms. These studies also have estimates provided by a single assay. We found parallel and sequential testing algorithms were potentially effective in ruling out false-positive cases given by RDTs (−7.8% in prevalence estimates using a single assay) and recognizing positive cases missed by ELISAs (+4.4%, Appendix A).

## 4. Discussion

In examining 1807 global serosurveys published between 1 January 2020 and 19 November 2021, we found that 192 unique commercial and 380 unique self-developed serological assays were used. A total of 50 commercial assays are used across 91% of SARS-CoV-2 seroprevalence studies. We found that intra-assay performance evaluations varied widely according to evaluation method and source. This variation in assay evaluations may have an impact on seroprevalence estimate validity and bias by under- or over-estimating estimates by up to 9.5%.

Serological assay performance is context dependent. Previous literature did not focus on assessing intra-assay consistency across different sources of validation for assays but put more emphasis on inter-assay comparisons [22,23,24,25,26]. Our study reveals that manufacturer evaluations of assay Sn. and Sp. were overestimations compared to independent and third-party head-to-head validations. Our pooled analysis found that Sn. on average was lower by 1.0% and 3.3% in third party and independent group evaluations, respectively. Likewise, Sp. on average was lower by 0.9% and 0.2% in third party and independent group evaluations, respectively. These results imply there may be more false positives and negatives than would be expected given manufacturer-verified test evaluations, which may impact result adjustment and interpretation.

### 4.1. Third-Party Evaluation Validates Manufacturer Data

Third-party evaluations are essential for more objective estimates of Sn. and Sp., enabling retrospective adjustment of seroprevalence data and selecting candidate assays for new studies. The five third-party labs included in our study all disclosed the reference panel they used (Appendix A). The composition of samples in reference panels is consistent across the evaluation of each individual assay, including testing materials consisting of combinations of high-titer, mid-titer, and low-titer samples on N- and S- antibody targets. Reference panels reflect the full-time course of infection (past infections, and waning antibodies). It also mirrors the complexity of antibody detection in real settings [27,28], as cross-reactivity to other viral infections (such as HIV, Dengue, Malaria, and Middle East Respiratory Syndrome) was also consistently assessed in negative panels. Third-party evaluations are of value in retrospectively adjusting data or selecting and adjusting for assays in new studies. However, these evaluations typically only target frequently used commercial ELISA and CLIA assays, which were less distributed in low-income regions like Africa (Appendix A).

### 4.2. Independent Evaluation Reflects Regional Population Characteristics

This situation necessitates that study investigators validate assays not included in these third-party evaluations. These independent evaluations better reflect the study geography, demographic context, epidemiological time course, and variant landscape, minimizing spectrum bias. Of note, studies have demonstrated loss of Sn. over time as antibodies wane, and incorporating performance based on time since the infection will gain further importance as the pandemic progresses [10,29]. Moreover, viral mutations may result in decreased assay performance [30]. Additionally, studies have shown differences in antibody dynamics in specific populations such as those from sub-Saharan Africa, young adults, and pregnant women that may impact test performance [31,32,33,34]. This step is not always feasible for all research settings, as we found only a small proportion (6.9%) of independent author groups conducted their own assay pre-study validation before rolling out their serosurvey. Fewer described the evaluation panels and methods they used. We encourage future studies to integrate assay evaluation more into a serosurvey design as a pre-step. Independent evaluations targeted toward the intended study population will update the understanding of serological assay performance, and conversely, accurate seroprevalence estimates.

### 4.3. Correct Seroprevalence Estimate for Assay Performance

Seroprevalence estimates can vary considerably based on the assay used, even in the same population and based on the same samples [23]. For instance, low-Sp. assays can lead to overinflated seroprevalence estimates, creating misleading results—particularly in settings with low true prevalence [35]. Moreover, Sn. and Sp. are not true parameters of the assays but can vary for the same assay depending on the reference panel or population used. Overall, our findings caution against accepting aggregate Sn. and Sp. reported by assay manufacturers, favoring independent or third-party evaluations on representative populations. Sn. and Sp. should be stratified by disease severity and time since infection, and the characteristics of the positive and negative reference panels should be reported at a minimum. The chance of biased estimates can be substantially minimized with proper adjustment. Our finding implies that statistically adjusting for test validity may be an essential step, particularly in low prevalence settings where a small absolute difference in seroprevalence can produce a massive relative difference in the understanding of case ascertainment, and/or where assay performance values are low (as seen with some rapid test assays).

### 4.4. Multiple Testing

Another option to minimize bias in seroprevalence studies was to use a multiple-testing strategy. Although findings should be further validated due to the heterogeneity of data, we noticed that pairing RDT with other assays could minimize false-positive rates by using RDTs only. RDT as a preliminary screening test suggests whether the test recipient produces any antibodies against SARS-CoV-2 in general. However, series confirmation tests identify the source of antibodies (infection/vaccination) and helps determine a much more precise timepoint of infection, see article [36]. Using just one RDT should be avoided. Moreover, multiple testing algorithms could also increase the Sn. of laboratory binding assays such as ELISAs and CLIAs and rule out false negatives, especially in low prevalence settings. Requiring a positive result on multiple assays in parallel and sequential testing improves the overall Sp. of the testing algorithm compared with the individual assays alone [37], but sometimes at the expense of Sn. [38]; conversely, requiring a positive result on just one of multiple assays improves Sn. at the expense of Sp. Sn. and Sp. should be taken as a whole to improve the positive predicted value of a testing algorithm to truly identify positive cases among all positive tests. Rational deployment of these algorithms should also consider contextual factors such as background prevalence in the population being studied, as positive predictive values are substantially lower in low prevalence settings [27]. Additionally, for accurate interpretation, reporting the details of the assays used and how they were combined with one another is important.

The combined Sn. and Sp. is calculatable for multiple testing algorithms based on individual performance features under either rule [36,39], but reporting a combined Sn. and Sp. at the point of completing all steps for a multiple testing algorithm on a regional sample is more preferable.

### 4.5. Limitations

This study has some limitations. While the living review from which our data is drawn captures all seroprevalence studies, we have not captured all applications of serological assays. For example, we excluded studies performed exclusively in confirmed COVID-19 cases and vaccinated individuals, and our findings may not apply to these areas of serological research. Additionally, our findings apply to population-based contexts and may not translate to the patient or clinical level, where serological assays are used to guide patient care. When collating third-party, independent, and manufacturer data on assay performance, we extracted the overall Sn./Sp. on total antibodies whenever available. We performed an empirical synthesis, making the best effort to collate all assay performance data accessible from online dashboards, preprints, institutional reports, and academic journals by identified sources. We extracted performance data collected from the far-most day from symptom onset. Finally, while we made our best effort to identify and summarize the use of serological assays in each serosurvey and the performance of assays from different sources, we saw people miss reporting performance matrices for self-developed assays. Therefore, we did not proceed with analyses for self-developed assays on performance comparison. Studies released as conference abstracts (48/1807, 2.7%) did not have enough space to describe the type of test used in a serosurvey in detail. However, given that the number of conference abstracts is small, it did not contribute a major result bias.

## 5. Conclusions

In conclusion, we found a large and diverse number of assays used in seroprevalence studies. This diverse selection of serological assays may impact the interpretation and reliability of seroprevalence estimates by up to ±9.5%, as Sn. and Sp. are not fixed properties of a serological assay but varying features depending on the reference panel or population on which is tested. We strongly recommend that: (1) authors conducting seroprevalence studies should consider adopting third-party or independently evaluated assays, which inform assay properties in a particular context; (2) statistical test adjustments on population seroprevalence should be employed using validated assay performance data; and (3) utilizing multiple testing strategies where possible (reporting a combined overall Sn. and Sp.) to minimize the risk of bias in seroprevalence estimates.

## Figures and Tables

**Figure 1 vaccines-10-02000-f001:**
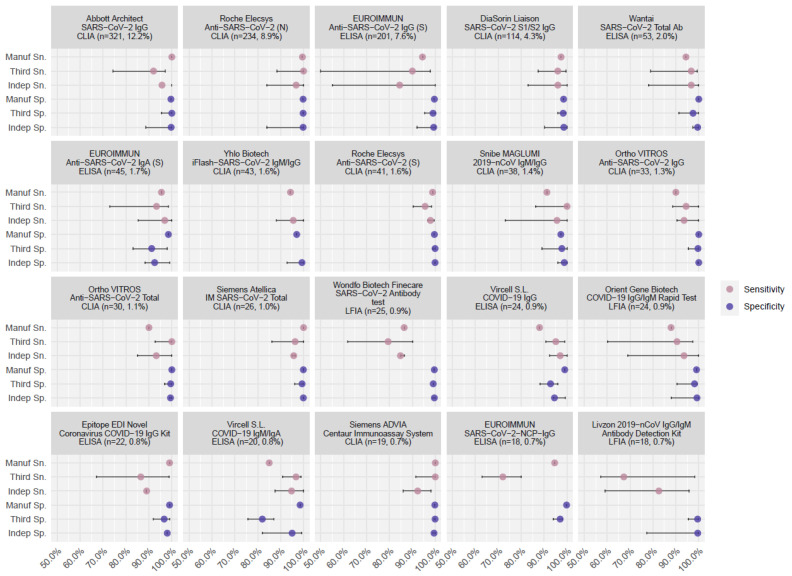
The difference in reported assay performance among manufacturer evaluation, third party evaluation, and independent evaluation. Note: the figure shows the side-by-side comparison of assay performance for the top 20 assays. Performance evaluations came from three sources: manufacturer reports, third-party labs, and independent groups. Intervals show the range of performance values for a certain assay derived from the given evaluation source.

**Figure 2 vaccines-10-02000-f002:**
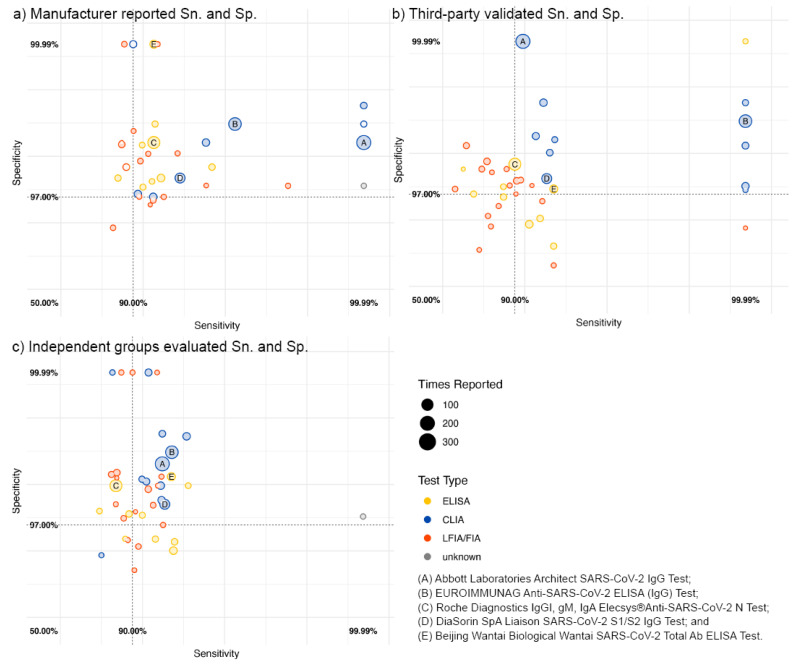
Sensitivity and specificity based on (**a**) manufacturer, (**b**) third-party, and (**c**) independent group evaluations for the top 50 most frequently used commercial serological assays. Note: both axes are on a log scale. Assays missing the corresponding source of evaluation were not involved in the analysis. The vertical and the horizontal lines indicate the WHO thresholds for Emergency Use Authorizations for COVID-19 serological assays: sensitivity minimum of 90.0%, and specificity minimum of 97.0%, respectively. Assays on the upper right area of each panel meet the WHO criteria for emergency use based on the dataset in question.

**Figure 3 vaccines-10-02000-f003:**
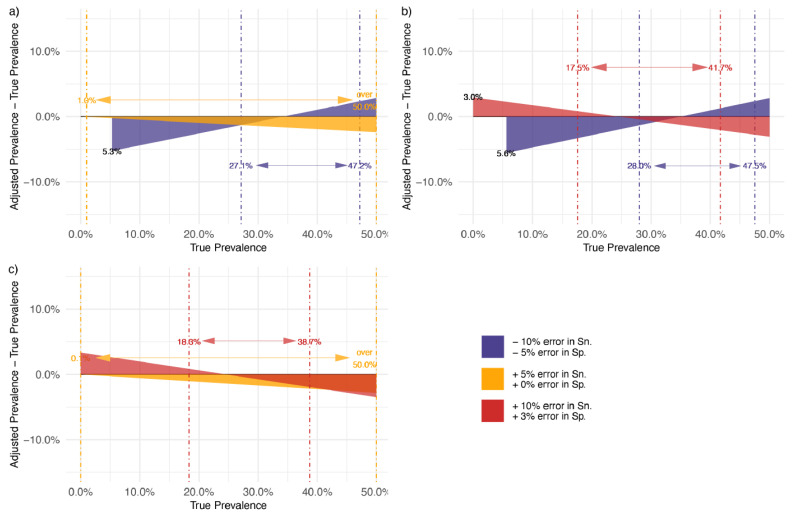
Consequences of correcting seroprevalence estimates using biased estimates of sensitivity (Sn.) and specificity (Sp.): simulation-based analysis. Serological assay with (**a**) high: a true Sn. at 95.0% and a true Sp. at 99.0%, (**b**) good: a true Sn. at 90.0% and a true Sp. at 97.0%, (**c**) and moderate performance: a true Sn. at 80.0% and a true Sp. at 87.0%. Note: The dot–dash lines provide an interval which indicates the seroprevalence adjusted for the mis-specified assay performance at a given error level was still within ±5% deviation of the true seroprevalence. The prevalence adjustment was performed using the formula by Sempos and Tian [20]. Notice that an assay with underestimated Sn. and Sp. is unable to provide prevalence estimates after adjustment at a low prevalence setting: (**a**) 5.3% and (**b**) 5.6%. An assay with overestimated Sn. and Sp. tends to inflate seroprevalence after adjustment when the true prevalence is low: (**b**) 3.0%.

**Table 1 vaccines-10-02000-t001:** Features of commercial and self-developed serology assays used by studies.

Assay Characteristics	Commercially Assays		Self-Developed Assays	
	(N = 192)		(N = 380)	
	n	%	n	%
Developed by				
*Manufacturer*	162		-	
*Lab groups*	-		275	
Type of Assays				
*ELISA*	60	31.3	261	68.7
*LFIA*	75	39.1	0	0.0
*IFA*	5	2.6	17	4.5
*CLIA (Including CGIA, CMIA)*	30	15.6	3	0.8
*Neutralization Assay*	0	0.0	52	13.7
*Others/Not specified*	22	11.5	47	12.4
WHO regions of development				
*Africa*	0	0.0	12	3.2
*America*	49	25.5	152	40.0
*Eastern Mediterranean*	5	2.6	15	3.9
*Europe*	75	39.1	163	42.9
*South East Asia*	4	2.1	6	1.6
*Western Pacific*	58	30.2	32	8.4
*Not Reported*	1	0.5	0	0.0
Feature of Assays				
*RDT*	103	53.6	17	4.5
*Non-RDT*	89	46.4	363	95.5
Antibody Targets				
*Spike*	55	28.6	48	12.6
*Nucleocapsid*	37	19.3	37	9.7
*Multiplex Targets* ^a^	38	19.8	171	45.0
*Unknown*	62	32.3	124	32.6
Isotypes				
*IgG-only*	52	27.1	149	39.2
*IgG and IgM*	103	53.6	31	8.2
*Total Antibody* *(IgG, IgM, IgA)*	22	11.5	38	10.0
*Other Combinations*^b^/ *Not Reported*	15	7.8	162	42.6
Assay Sn. and Sp.				
Manufacturer/developer reported	91	47.4	124	32.6
Third-party validated	118	61.5	-	-
*Australia NRL*	16	8.3	-	-
*Australia Doherty*	18	9.4	-	-
*US FDA*	57	29.7	-	-
*FIND Diagnostic*	30	15.6	-	-
*Netherland CIDC*	26	13.5	-	-
*Other groups*	94	49.0	-	-
Emergency Use ^c^				
*Yes*	57	29.7	-	-
*No*	135	70.3	-	-

Note: ^a^ Multiplex targets indicate the assay detects more than one targets on the SARS-CoV-2 virus; the multiplex detection combinations include spike-whole virus antigen, spike-nucleocapsid, nucleocapsid-spike-envelope protein. ^b^ Other test isotypes include IgM-only, IgA-only, IgM and IgA, and other not-specified isotype combinations. ^c^ Manufacturer reported test performance met the WHO standards for Emergency Use Authorizations for COVID-19 serological tests: sensitivity minimum 90.0%, specificity minimum 97.0%.

**Table 2 vaccines-10-02000-t002:** Predictors of assay Sn. and Sp. estimated with mixed-effect beta regression (N = 192).

Fixed Effects	Sensitivity				Specificity			
	Difference in Performance against Manufacturer Value ^a^	Absolute Performance Value ^b^	*p* ^c^		Difference in Performance against Manufacturer Value ^b^	Absolute Performance Value ^a^	*p* ^c^	
	[95% CI]	[95% CI]			[95% CI]	[95% CI]		
Source of Evaluation								
*Manufacturer*	ref.	93.6%	<0.001	*	ref.	98.5%	<0.001	*
		[90.6, 95.7%]				[97.8, 99.0%]		
*Independent*	3.3%	90.3%	0.001	*	0.2%	98.3%	0.247	
	[2.7, 3.4%]	[87.8, 92.3%]			[−0.1, 0.4%]	[97.8, 98.7%]		
*Third Party’s Lab*	1.0%	92.6%	0.289		0.9%	97.6%	<0.001	*
	[0.1, 1.4%]	[90.5, 94.3%]			[0.9, 0.9%]	[96.9, 98.2%]		
*NRL*	−2.2%	95.8%	0.207	*	4.2%	94.4%	<0.001	*
	[−2.3, −1.8%]	[92.9, 97.5%]			[2.7, 6.4%]	[91.3, 96.4%]		
*US FDA*	−2.2%	95.8%	0.038	*	0.4%	98.1%	0.047	*
	[−3.6, −1.3%]	[94.2, 97.0%]			[0.4, 0.4%]	[97.3, 98.6%]		
*FIND Diagnostic*	18.6%	75.0%	<0.001	*	0.9%	97.6%	0.008	*
	[14.6, 22.8%]	[67.8, 81.1%]			[0.6, 1.3%]	[96.4, 98.4%]		
*Netherland CIDC*	−0.2%	93.8%	0.825		0.5%	98.0%	0.060	
	[−0.3, 0.0%]	[90.5, 96.0%]			[0.4, 0.7%]	[97.0, 98.7%]		
*Doherty*	2.7%	90.9%	0.055		0.8%	97.7%	0.037	*
	[1.5, 4.5%]	[86.1, 94.1%]			[0.4, 1.5%]	[96.2, 98.6%]		

Note: Assay features of isotype, test type, antibody targets, multiplex detection, and time to result were fitted as covariates to adjust outputs. ^a^ Coefficients derived from the model were in log-odds formats, which were then converted to percentages for absolute performance value estimates. ^b^ Difference in performance matrix against manufacturer value was estimated using bootstrapping with 10,000 iterations. ^c^ corresponds to the significant level of *p* < 0.05; *p* attaches to absolute performance values. The mixed beta regression model bounded between 0–100%, clustering assay performance data in serological assays groups. * corresponds to the significant level of *p* < 0.05.

## Data Availability

Data from seroprevalence studies and the serological assays used therein are available from: https://serotracker.com/en/Explore (accessed on 20 November 2022).

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
