# Peer review of "Serology Assays Used in SARS-CoV-2 Seroprevalence Surveys Worldwide: A Systematic Review and Meta-Analysis of Assay Features, Testing Algorithms, and Performance"

_vaccines, 2022, doi:10.3390/vaccines10122000_

Round 1

Reviewer 1 Report

The authors carried out a thorough literature analysis on the performance and use of SARS-CoV-2 seroprevalence studies and the assays utilized by them. Following a systematic screen, they provide the comparative characteristics of the assays, discuss the reporting of assay performance and the effects these reports have on seroprevalence estimations. Overall, the breadth and quality of analysis is convincing and the reported results can improve SARS-CoV-2 seroprevalence estimates in the future.

I have only minor remarks.

Figure 3 There is a reference to a paper by Sempos and Tian, please use the number of the citation (20) and not the abbreviation of the journal. The number of colors and shades are difficult to interpret in this figure, perhaps the display of only High&Moderate performance assay effects would be easier to interpret and code in colors.

Supplementary figure S3 contains information that, I think, deserves to be included in the main part of the paper.

The key recommendations of the authors could be collected and listed as highlights or in a table, this would help the readers remember the take-home messages of the analysis.

Author Response

The authors carried out a thorough literature analysis on the performance and use of SARS-CoV-2 seroprevalence studies and the assays utilized by them. Following a systematic screen, they provide the comparative characteristics of the assays, discuss the reporting of assay performance and the effects these reports have on seroprevalence estimations. Overall, the breadth and quality of analysis is convincing and the reported results can improve SARS-CoV-2 seroprevalence estimates in the future.

Response: We appreciate your comments on this paper. Please find our point-to-point response to each of the comments below.

I have only minor remarks.

Figure 3 There is a reference to a paper by Sempos and Tian, please use the number of the citation (20) and not the abbreviation of the journal. The number of colors and shades are difficult to interpret in this figure, perhaps the display of only High&Moderate performance assay effects would be easier to interpret and code in colors.

Response: Thank you for your recommendation. We replaced the journal name with a citation number in the legend of Figure 3. To improve visualization in Figure 3, we decreased the number of ribbons from 3 to 2 in each subfigure. We hope this change can improve interpretability.

Supplementary figure S3 contains information that, I think, deserves to be included in the main part of the paper.

Response: Thank you for your recommendation. We also have had a hard time deciding between Figure 2 and Figure S3. As suggested, we have moved Figure S3 to the main part, moving Figure 2 to the supplement material close to the pairwise third-party evaluation comparison figure.

The key recommendations of the authors could be collected and listed as highlights or in a table, this would help the readers remember the take-home messages of the analysis.

Response: Thank you for your comment. Given that the space is limited, and Vaccines do not encourage formatting the paper with a highlight box at the front page, we collated take-home messages as bullet points in the conclusion (page 18, lines 378-388).

Reviewer 2 Report

The submission contains some important information and therefore I am supportive for its publication after the major revision.

Here are the major comments:

Major Comment 1: mixed beta regression model shall be using correlated errors, or serious justification for standard structure shall be given.

Major Comment 2: I suggest to sharpen both Abstract and conclusions. These are too tersely written. 

Major comment 3: It was indeed observed, that the use of only one test with specificity below 100%

could be seriously imprecise,  can already be read in the articles like  R. Mead (1992) . This motivates the practical idea of practically combining at least two or more   tests. Read the reference and incorporate in your Sp. and Sn. differences between manufacturer and independent checks.

Roger Mead, University of Reading, England, Statistical Games 2 - Medical Diagnosis,  Issue 3 p. 12-16 Teaching Statistics Volume 14. Number 3, Autumn 1992

Major comment 4: page 3, Data Sources: How you guarantee that your source collection is not biased? Justification is needed.

Major comment 5: it is indeed a serious limitation not to have knowledge about descriptions of tests used (last sentence of discussion, p.12). Please, write on this limitation also in abstract and conclusion explicitly, or avoid usage of such abstracts.  

Author Response

Major Comment 1: mixed beta regression model shall be using correlated errors, or serious justification for standard structure shall be given.

Response: Thank you for your comment. Previously we ran the mixed beta regression model without specifying structured variance-covariance matrices for the sensitivity and the specificity models. In this update, we compared the estimates using different VCOV matrices including diagonal heterogeneous matrix structure, compound symmetric matrix structure, and Toeplitz matrix structure. ANOVA test turned out that with very similar estimated values given, models using different correlation structures did show significant differences in terms of fitness and coefficient estimates. Given the high heterogeneity of source data, we decide to stay with using the diagonal heterogeneous variance-covariance structure. One variable was also added to the new model to improve the fitness of the model (old vs new sensitivity model: p = 0.049 [ANOVA test logit model], old vs new specificity model: p = 0.017 [ANOVA test for logit model]). The final reported numeric values were slightly different from the previous model, but the conclusion for this study remains the same. In the new modeling table, we removed covariate estimates because the values are hard to interpret and added an estimated difference in performance value against manufacturer values. Additional modeling details and justification as above were added to the main text (page 9, lines 172-182).

“In examining whether assay performance differs by evaluation sources, we developed separate mixed-effect beta regression models for Sn. and Sp. with random effects specified for individual serological assays. Given that data with high heterogeneity, a diagonal heterogeneous variance-covariance structure was finally selected when estimating the assay performance by evaluation source. Assay features of isotype, test type, antibody targets, multiplex detection, and time to result were fitted as covariates to adjust outputs. Raw log odds obtained from models were converted to percentage for ease of interpretation. Difference in performance matrix against manufacturer values with 95% Confidence Interval (95% CI) by evaluation sources was derived using bootstrapping with 10000 iterations.”

Major Comment 2: I suggest to sharpen both Abstract and conclusions. These are too tersely written.

Response: As suggested, we have revised the abstract and conclusions (page 18). The discussion overall has also been streamlined as well (pages 13-17).

Major comment 3: It was indeed observed, that the use of only one test with specificity below 100% could be seriously imprecise,  can already be read in the articles like  R. Mead (1992) . This motivates the practical idea of practically combining at least two or more tests. Read the reference and incorporate in your Sp. and Sn. differences between manufacturer and independent checks.

Roger Mead, University of Reading, England, Statistical Games 2 - Medical Diagnosis,  Issue 3 p. 12-16 Teaching Statistics Volume 14. Number 3, Autumn 1992

Response: Thank you for sharing this reference with us. We agree that the measure of sensitivity and specificity described by R. Mead is especially important in settings like treating COVID-19 where identifying true positive cases is necessary and where handling a false positive is extremely costly. We have added the reference and relevant reflections into the text, and also did an additional search on studies for the combined use of multiple diagnostic tests. The discussion was expanded accordingly (page 16-17, line 341-357).

“Rational deployment of these algorithms should also consider contextual factors such as background prevalence in the population being studied, as positive predictive values are substantially lower in low prevalence settings[27]. Additionally, for accurate interpretation, reporting the details of the assays used and how they were combined with one another is important. The combined Sn. and Sp. is calculatable for multiple testing algorithms based on individual performance features under either rule[38], but ideally reporting a combined Sn. and Sp. after going through all testing steps on a regional sample is preferable.”

Major comment 4: page 3, Data Sources: How you guarantee that your source collection is not biased? Justification is needed.

Response: Thank you for your comment. While publication bias is an ever-present issue in systematic review study design, we attempted to mitigate bias in our source collection by including source types beyond peer-reviewed articles such as preprints, institutional reports, media reports, and grey literature. See the following text modified to the methods section, paragraph 2, page X: “We developed our published literature, preprint, and grey literature search strategies with a health sciences librarian, and attempted to mitigate possible publication bias by including a variety of sources beyond peer-reviewed articles such as preprints, institutional reports, media sources, and grey literature” We also added description in the limitation section on the potential bias of  source data collection (page 17, line 370-376).

Major comment 5: it is indeed a serious limitation not to have knowledge about descriptions of tests used (last sentence of discussion, p.12). Please, write on this limitation also in abstract and conclusion explicitly, or avoid usage of such abstracts. 

Response: We acknowledge that this is indeed a limitation and we have added a short mention of it into our abstract and conclusion as suggested. We elaborated more detail on this limitation in our discussion, as we believe this limitation is not a major source of flaws in the validity of results. For full journal articles, the testing method is a necessary part of to report according to the article publication checklist. For commercial assays described in a journal article and institutional reports, we added a sentence in the manuscript “We used the information of manufacturer name, assay name, assay type, antibody target, and Catalog number (if available), and cited source articles that describes the assay development to ensure the serological assay mapping was not ambiguous.” For self-developed assays, we added a paragraph in the method stating that “Evaluation data was not very available for self-developed assays as for commercial assays, in the assay description of which concentrated on the steps of developing such an assay with performance matrices provided randomly. Therefore, corresponding performance analysis was not conducted for self-developed assays.

Even though conference abstracts provided very limited data on assays used for either self-developed assays or commercial assays, there were only 48 out of 1807 articles can be categorized into this type. We have rephrased the last sentence of the description. We highlighted this point in the limitation section (page 17, lines 370-376). We also highlighted in the abstract that “More investigation should be directed to consolidating performance of self-developed assays.

Round 2

Reviewer 2 Report

Authors did some job in revision. But unfortunately, current revision is still not satisfactory.

Here are the major comments:

Addendum to Major comment 3: Reference to   R. Mead (1992) . is still missing in the manuscript. Where it is cited?

Addendum to Major comment 4: page 3, Data Sources: More quality justification is needed. How you guarantee that your source collection is not biased? 

Author Response

Addendum to Major comment 3: Reference to   R. Mead (1992). is still missing in the manuscript. Where it is cited?

Responses: Our apologies for not including this valuable reference source in the last round. We replaced it with another reference we recommended by one of our authors. In this latest manuscript, we have cited “Statistical Games 2 - Medical Diagnosis” in the parts that discuss identifying antibody sources and deriving combined sensitivity and specificity.

“we noticed that pairing RDT with other assays could minimize false-positive rates by using RDTs only. RDT as a preliminary screening test suggests whether the test recipient produce any antibodies against SARS-CoV-2 in general. The series confirmation tests following identifies the source of antibodies (infection/ vaccination) and helps determine a more precise timepoint of infection[36].”

“The combined Sn. and Sp. is calculatable for multiple testing algorithms based on individual performance features under either rule[36, 39], but reporting a combined Sn. and Sp. at the point of completing all steps for a multiple testing algorithm on a regional sample is more preferable.”

Addendum to Major comment 4: page 3, Data Sources: More quality justification is needed. How do you guarantee that your source collection is not biased? 

Responses: Thank you for your comment. We acknowledge that publication bias is an ever-present issue in systematic review study design. We attempted to mitigate bias in our source collection by including source types beyond peer-reviewed. We expanded on the discussion of search strategies for reducing search bias in the text: To mitigate bias, we created a search strategy that was as thorough as possible in comprehending the immunoassays utilized in seroprevalence studies. All identified articles were recorded in the SeroTracker database, a database containing the most comprehensive source of seroprevalence research ever made available. The search strategy identified published literature, and preprints was created in collaboration with a health sciences librarian. We sought to reduce any potential publishing bias by adding a range of sources besides peer-reviewed publications, including institutional reports, media sources, and grey literature. Experts who collaborated with us and the users of the SeroTracker website recommended grey literature.”  We have elaborated on the data collection steps for commercial assay and self-developed assay from Line 121, Page 7.

We admitted the limitation that: “While the living review from which our data is drawn captures all seroprevalence studies, we have not captured all applications of serological assays. For example, we excluded studies done exclusively in confirmed COVID-19 cases and vaccinated individuals, and our findings may not apply to these areas of serological research.” As have known this fact, we highlighted that our recommendations for serological assay usage in the future only apply to research that aims to understand population seroprevalence. In the conclusion we suggested that “1) authors conducting seroprevalence studies should consider adopting third-party or independently evaluated assays, which inform assay properties in a particular context; 2) statistical test adjustments on population seroprevalence should be employed using validated assay performance data, and 3) utilizing multiple testing strategies where possible (reporting a combined overall Sn. and Sp.) to minimize the risk of bias in seroprevalence estimates.” We hope this alleviates your concerns about potential bias in data collection and any potential inappropriate extrapolation of study findings.

Round 3

Reviewer 2 Report

Authors did job in revision. I am prone to recommend publication of the manuscript after better editing of text related to my Addendum to Major comment 3.

I kindly but importantly request to better style the text, here is my recommendation: 

“we noticed that pairing RDT with other assays will minimize false-positive rates by using RDTs only. RDT as a preliminary screening test suggests whether the test recipient produce any antibodies against SARS-CoV-2 in general. However,   series confirmation tests following identifies the source of antibodies (infection/ vaccination) and helps determine a much more precise timepoint of infection, see [36]. Using just one RDT should be avoided.”

“The combined Sn. and Sp. is calculatable for multiple testing algorithms based on individual performance features under either rule[36, 39], but reporting a combined Sn. and Sp. at the point of completing all steps for a multiple testing algorithm on a regional sample is more preferable.”

After using this text I am recommending publication of the manuscript. Sincerely, Referee

Author Response

We appreciate that the reviewer went over our manuscript again in great detail. We have used the suggested text in the latest revision, which was highlighted in yellow in the document.